# Comparative Analysis of Volatile Terpenes and Terpenoids in the Leaves of *Pinus* Species—A Potentially Abundant Renewable Resource

**DOI:** 10.3390/molecules26175244

**Published:** 2021-08-29

**Authors:** Wensu Ji, Xiaoyue Ji

**Affiliations:** 1Ordnance Non-Commissioned Officers School, Army Engineering University of PLA, Wuhan 430075, China; wensuji324@163.com; 2Advanced Analysis & Testing Center, Nanjing Forestry University, Nanjing 210037, China

**Keywords:** volatile terpenes and terpenoids, pine leaves resource, principal component analysis, cluster analysis, biodiversity

## Abstract

*Pinaceae* plants are widely distributed in the world, and the resources of pine leaves are abundant. In the extensive literature concerning *Pinus* species, there is much data on the composition and the content of essential oil of leaves. Still, a detailed comparative analysis of volatile terpenes and terpenoids between different species is missing. In this paper, headspace solid-phase microextraction coupled with gas chromatography-mass spectrometry was used to determine the volatile terpenes and terpenoids of typical *Pinus* species in China. A total of 46 volatile terpenes and terpenoids were identified, and 12 common compounds were found, which exhibited a great diversity in the leaves of *Pinus* species. According to the structures and properties of the compounds, all those compounds can be classified into four categories, namely monoterpenes, oxygenated terpenes, terpene esters, and sesquiterpenes. The results of principal component analysis and cluster analysis showed that the leaves of the six *Pinus* species could be divided into two groups. The species and contents of volatile terpenes and terpenoids in the leaves were quite different. The results not only provide a reference for the utilization of pine leaves resource, but also bring a broader vision on the biodiversity.

## 1. Introduction

*Pinus* species are widely distributed in the world. Due to their strong adaptability, *Pinus* species are indispensable tree species in afforestation. They are also a critical timber resource in the forest market [1]. Besides, *Pinus* species exhibit tremendous exploitation potential for medicinal application [2]. Specifically, the leaves of *Pinus* species, commonly called pine needles, are an important part of the *Pinus* species due to their medicinal use. Their medicinal components are higher than other parts of *Pinus* species. Phytochemistry research has shown that the pine leaves extract is rich in carbohydrate, crude protein, crude fat, various amino acids, vitamins, bimetallism, essential oil, chlorophyll, unsaturated fatty acids, enzymes and coenzymes, and other active substances [3,4]. Modern pharmacological studies showed that pine needle extract has analgesic, anti-inflammatory, antipruritic, sedative, antitussive, expectorant, antiasthmatic, lipid regulating, antioxidant, antimutation, and antitumor bioactivities [5,6,7]. Considering that pine leaves are also very abundant in the world, it is of great significance to develop and utilize this renewable resource in depth.

Forest trees are the primary source of volatile organic compounds (VOCs), which constitute a large and diverse category of plant secondary metabolites [8]. The main function of these compounds is to protect plants from herbivores and pathogenic microorganisms [9,10,11]. VOCs emitted by plants are mainly terpenes and terpenoids, with about 55,000 different structures [12]. The structures of terpenes are based on the linkage of isoprene unit (C_5_H_8_). Two or more C_5_ components generate various terpenes through a head–tail condensation reaction [13]. Different from terpenes that are hydrocarbons, terpenes containing additional functional groups, usually oxygen-containing, are called terpenoids [14]. Terpenes and terpenoids have different biological activities [15]. Among them, the beneficial effects of terpenes and terpenoids on human health have been concerned by many researchers [16,17]. Their roles in various human diseases, such as inflammatory diseases, tumorigenesis and neurodegeneration, have been studied by cell and animal models for decades, suggesting that terpenes and terpenoids are potential chemopreventive and therapeutic drugs for various diseases [18]. In recent years, some new terpenoids have been isolated or synthesized, providing more terpenoids with potential chemotherapeutic value for clinical research [19].

At present, due to the importance of pine leaves, a large number of studies have reported on the volatile oil of pine leaves, but mainly on the chemical composition of a single kind of pine leaves [20,21]. There are few reports on the comparative study of volatile oils of different pine plants, and the comparative analysis of volatile terpenes and terpenoids in different pine leaves is missing. In this study, volatile terpenes and terpenoids of representative *Pinus* species (*Pinus sylvestris*, *Pinus tabuliformis, Pinus bungeana, Cedrus deodara, Pinus densiflora, Pinus thunbergii*) in China were identified using headspace solid-phase microextraction coupled with gas chromatography-mass spectrometry (HS-SPME-GC-MS). Through a practical principal component analysis (PCA) and cluster analysis (CA), differences of volatile terpenes and terpenoids among different pine leaves were compared. The obtained results will fill the research gap in the comparative analysis of volatile terpenes and terpenoids obtained from the leaves of *Pinus* species, as well as may provide the reference for the resource utilization of *Pinus* species leaves. Furthermore, this study could provide more knowledge for understanding biodiversity.

## 2. Results and Discussion

### 2.1. Identification and Comparison of Volatile Terpenes and Terpenoids

HS-SPME-GC-MS was used to detect volatile terpenes and terpenoids in six typical pine leaves. The compounds were qualitatively analyzed by searching the spectrum library (Nist14) and combining it with the retention index (RI) of the compounds. The results showed that 46 volatile terpenes and terpenoids were detected in the leaves of six *Pinaceae* species, including 12 of the same components. According to the structure of compounds, all compounds can be divided into four categories, namely monoterpenes (15), oxygenated terpenes (7), terpene esters (4), and sesquiterpenes (20). Identification and classification of the volatile terpenes and terpenoids were reported in Table 1.

These compounds usually have important biological activities. Among them, monoterpenes are mainly used for analgesia, sterilization, antivirus, and sedation [22]. á-Pinene, and á-myrcene have analgesic, anti-inflammatory, antitussive and expectorant effects [23]. á-Pinene also has an obvious bactericidal and bacteriostatic effect on *Candida albicans*, and can hinder the deposition of blood lipid in the aorta and heart, as well as prevent atherosclerosis [24]. Limonene has the functions of sterilization, insect repellent, sedative central nervous system, expectorant, cough, antiasthmatic, and gallstone dissolution [25]. Terpinolene and camphene have anthelmintic and antibacterial effects, among which terpinolene can also prevent atherosclerosis [26,27].

Sesquiterpenes usually have the effects of lowering blood pressure, sterilization, anti-inflammatory, and analgesic [28]. For example, caryophyllene has antiasthmatic, antitussive, expectorant, and anti-inflammatory effects. It is clinically used for the treatment of tracheitis. It also shows the sedative effect on intense emotions [29]. Perfume with caryophyllene as the main component has been used in some obstetrics and gynecology hospitals to treat the puerpera’s irritability after childbirth. Besides, caryophyllene also has antitumor activity [30]. Copaene, germacrene D, and α-cadinene can resist insects, bacteria, and fungi [31].

Oxygenated terpenes and terpene esters also have noticeable health care effects, with strong bactericidal, anti-infection, antiviral, and stimulative effects [32]. For example, bornyl acetate has the ability to inhibit pulmonary inflammatory response [33] and inhibit the proliferation of many kinds of cancer cells [34]. It has an anti-inflammatory effect in human chondrocytes [35]. Linaool can slow down the respiratory rate, calm the nerves, reduce blood pressure, and has a good effect on the cardiovascular system and central nervous system [36].

### 2.2. Comparison of Volatile Terpenes and Terpenoids

Volatile terpenes and terpenoids exhibited a great diversity in the leaves of *Pinus* species. The analysis results indicated that the fraction of volatile terpenes and terpenoids in the leaves of *Pinus* species were different (Figure 1). Similar results are documented using the SPME or other pretreatment techniques. For example, Zorica S. Mitić et al. reported that the chemical composition of dominant terpenes of each Pinus species essential oil are different, and the essential oil is generally dominated by monoterpenes [37]. Tayyebeh Ghaffari et al. pointed out that the main component of pine needle essential oil is sesquiterpene grown in Northwestern Iran [38].

In general terms, among the six kinds of pine leaves, sesquiterpenes (20 compounds) were the most abundant, followed by monoterpenes and oxygenated terpenes (15 and 7 compounds, respectively), and terpene esters (4 compounds) were the least. According to the chromatogram, the compound’s content is directly proportional to the peak area, so we use the peak area to compare the content of the compound. It can be clearly seen from Figure 1 that the content of volatile terpenes and terpenoid is the highest in *P. densiflora* and the lowest in *C. deodara*. In *P*. *tabuliformis*, and *P. bungeana*, the number of volatile terpenes and terpenoids identified were the largest (27 compounds). Contrastively, the number of volatile terpenes and terpenoids detected was the least in the *P. sylvestris* (23 compounds) and *P. thunbergii* (23 compounds).

Concretely, the number of compounds of monoterpenes, oxygenated terpenes, terpene esters, and sesquiterpenes in the leaves of these six *Pinus* species were directly compared through Figure 2. Firstly, the content and number of monoterpenes compounds were the most in *P. thunbergii*, and the least *C. deodara*. Secondly, the content of oxygenated terpenes was the highest in *P. thunbergii* and the lowest in *P. bungeara*. The number of oxygenated terpenes compounds was the highest in *C. deodara* and *P. densiflora*. No oxygenated terpenes compounds were detected in sample *P. bungeara*. Interestingly, the content and number of terpene esters was the highest in *P. bungeara*. Finally, the content of sesquiterpenes compounds were the most in *P. densiflora*, and the least in sample *C. deodara*. The number of oxygenated terpenes compounds was the highest in *P. thunbergii*.

### 2.3. PCA of SPME-GC-MS

PCA is classic feature extraction and dimension reduction technology, which can be employed to simplify and optimize a large number of data [39]. In this experiment, factor analysis in SPSS software was used, and volatile terpenes and terpenoids in six kinds of pine leaves were used as variables to conduct PCA on six kinds of pine leaves. The results were shown in Figure 3.

The results show that the cumulative variance contribution rate of principal component 1 (66.577%), principal component 2 (22.293%), and principal component 3 (5.231%) in PCA analysis is 94.101%, which can reflect most of the information of the original data of the sample, indicating that PCA can better distinguish pine leaf samples. According to the results of PCA, the six pine leaves tested can be obviously divided into two categories (Figure 3). The volatile terpenes and terpenoids of pine leaves in *P. sylvestris*, *P. tabuliformis*, and *P. bungeana* are similar and have the same resource utilization, so that they can be classified into one category. The volatile terpenes and terpenoids in samples *C. deodara*, *P. densiflora*, and *P. thunbergii* were obviously different from the other three species. They were grouped into one group, which could be treated differently in resource utilization.

### 2.4. CA of HS-SPME-GC-MS

CA is a multivariate statistical analysis method. It compares the properties of each classification object, where classifies the objects with similar properties into one category, and classifies the objects with different properties into different categories. According to the difference of sample similarity measure or the principle of classification, CA’s means are various [35]. In this study, the Euclidean distance matrix was calculated based on the relative content of volatile terpenes and terpenoids in various pine leaves, and the CA of volatile terpenes and terpenoids was carried out by the complete linkage method. The results were expressed in the form of a thermal graph, which more intuitively reflected the distribution of volatile terpenes and terpenoids in these six kinds of pine leaves. The results were shown in Figure 4.

In Figure 4, the abscissa represents six pine needle samples, and the ordinate represents the volatile terpenes and terpenoids. The color of the thermodynamic diagram represents the relative content of volatile terpenes and terpenoids. The results showed that the results of CA were consistent with that of PCA. *P. sylvestris*, *P. tabuliformis*, and *P. bungeana* were clustered into one group, and the *C. deodara*, *P. densiflora*, and *P. thunbergii* were clustered into one group. In addition, it can be clearly seen from the thermodynamic diagram that the volatile terpenes and terpenoids in each group are different. Specifically, the relative contents of monoterpenes and sesquiterpenes varied greatly. Oxygenated terpenes and terpene esters had little difference. The results provided a reference for the utilization of pine needle resources.

## 3. Materials and Methods

### 3.1. Plant Material and Reagents

All fresh leaves of the plant were collected during April 2021. *P. sylvestris, P. tabuliformis, P. bungeana, C. deodara, P. densiflora,* and *P. thunbergii* were all collected from the campus of Nanjing Forestry University (32°54′ N, 118°50′ E). The samples were taken to the laboratory for analysis immediately after collection. *N*-alkane C8–C40 was purchased from Shanghai Jieli Biotechnology Co., Ltd. (Shanghai, China).

### 3.2. Instruments and Equipments

Trace ISQ gas chromatography-mass spectrometry (Thermo Fisher Scientific, Waltham, MA, USA); Manual SPME holder, 65 μm PDMS/DVB SPME fiber (Supelco, Bellefonte, PA, USA); HH.S21-6 electrothermal constant temperature water bath pot (Shanghai Boxun Industrial Co., Ltd. Medical Equipment Factory, Shanghai, China); BSA224s electronic balance (maximum weighing 220 g, minimum weighing 0.0001 g, error 0.1 mg) (Sartorius, Göttingen, Germany).

### 3.3. Test Method

#### 3.3.1. Sample Preparation and Extraction Process for HS-SPME

0.5G sample was accurately weighed and put into a 15 mL sample bottle sealed with a sealing film, and then the fiber was inserted into the bottle (the new fiber needs to be aged at the GC inlet according to the instructions before use). After 30 min of headspace adsorption 40 °C, it was pulled out and then inserted immediately into the GC/MS instrument with the inlet temperature of 250 °C. The desorption time was 3 min for GC/MS analysis. All experiments were repeated three times.

#### 3.3.2. GC/MS Analysis Conditions

Chromatographic conditions: DB-5MS capillary column (30 m × 0.25 mm, 0.25 μm film thickness) was employed. High purity helium (99.999%) was used as carrier gas with the flow rate of 1.0 mL/min. The inlet temperature was adjusted at 250 °C. The solvent delay time was 1 min. The injection mode was the nonreversible injection. The starting column temperature was 50 °C held for 3 min, programmed to 80 °C at 10 °C/min, then rising to 100 °C at 3 °C/min. The oven temperature was ramped 160 °C at 10 °C/min, then ramped to 180 °C at 3 °C/min, and finally ramped to 220 °C at 15 °C/min maintaining it for 2 min.

Mass spectrometry conditions: the ionization mode was electron ionization source (EI), and the electron energy was set at 70 eV. The transfer line temperature and ion source temperature were both set at 250 °C. Full scan mode was employed with a scanning quality range of 33–450 amu. The operating system was Xcalibur (v. 2.2, Thermo Fisher Scientific).

#### 3.3.3. Statistical Analysis

Volatile terpenes and terpenoids from pine leaves were mainly determined by two methods: (1) depending on MS Library search: the volatile terpenes and terpenoids were identified by comparing with the MS literature data (NIST 14); (2) depending on retention index (RI): homologous series of n-alkanes (C8–C40) was injected under the same chromatographic conditions [40]. The RI was calculated by the retention time of compounds to identify volatile terpenes and terpenoids.

Quantitative analysis of the percentage of each volatile terpene and terpenoid was calculated by the peak area normalization measurements.

## 4. Conclusions

For the first time, volatile terpenes and terpenoids of six representative *Pinus* species in China were comparatively analyzed. A total of 46 volatile terpenes and terpenoids were identified by HS-SPME-GC-MS. The most diverse compounds were sesquiterpenes with 20 compounds, followed by monoterpenes and oxygenated terpenes with 15 and 7 compounds, respectively. Terpene esters were the least diverse with only four compounds. Most of these volatile terpenes and terpenoids have essential biological activities. In addition, the volatile terpenes and terpenoids in the six pine leaves were analyzed by PCA and CA, and the pine leaves could be divided into two categories. The volatile terpenes and terpenoids in samples *C. deodara*, *P. densiflora*, and *P. thunbergii* were similar in terms of composition and content of compounds, so they were clustered into one category. Other samples (*P. sylvestris*, *P. tabuliformis*, and *P. bungeana*) were clustered into one category. This study may provide the reference for the resource utilization of *Pinus* species leaves. Further, this study could provide further knowledge for understanding biodiversity.

## Figures and Tables

**Figure 1 molecules-26-05244-f001:**
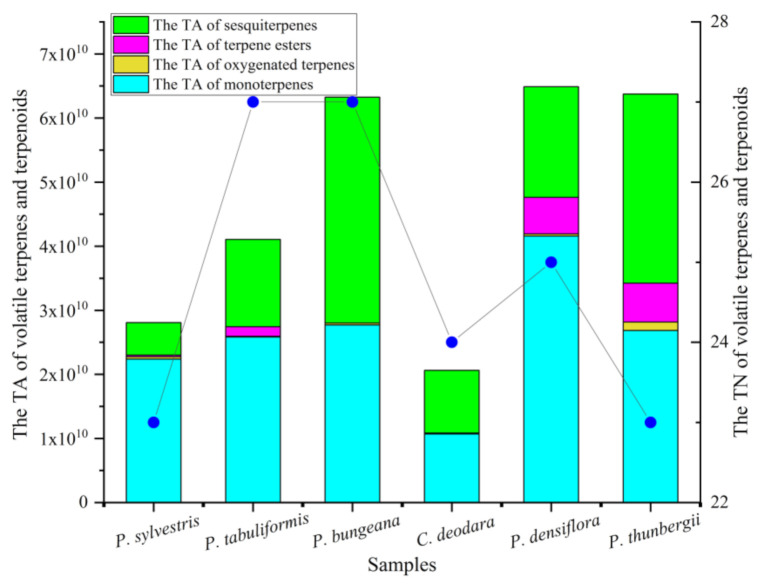
Comparison of volatile terpenes and terpenoids in pine leaves. TA refers to total area. TN refers to total number. The blue dots refers to the TN of volatile terpenes and terpenoids.

**Figure 2 molecules-26-05244-f002:**
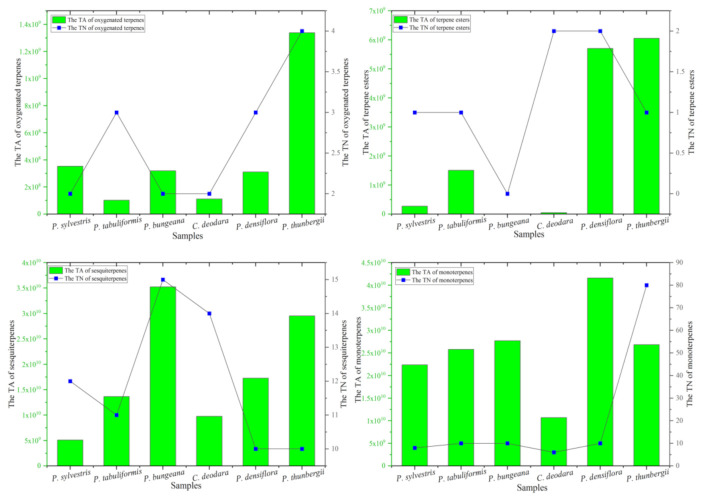
Comparison of monoterpenes, oxygenated terpenes, terpene esters, and sesquiterpenes in pine leaves. TA refers to total area. TN refers to total number.

**Figure 3 molecules-26-05244-f003:**
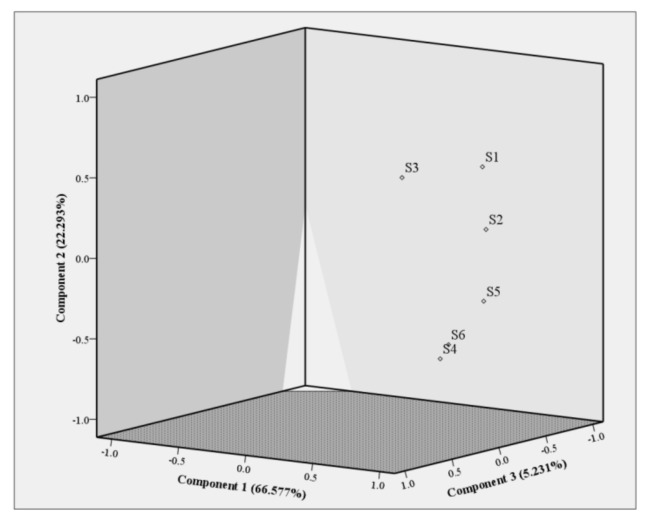
PCA plot of the pine leaf samples. S1–S6 represented *P. sylvestris*, *P.*
*tabuliformis*, *P. bungeana*, *C. deodara*, *P. densiflora*, and *P. thunbergii* respectively.

**Figure 4 molecules-26-05244-f004:**
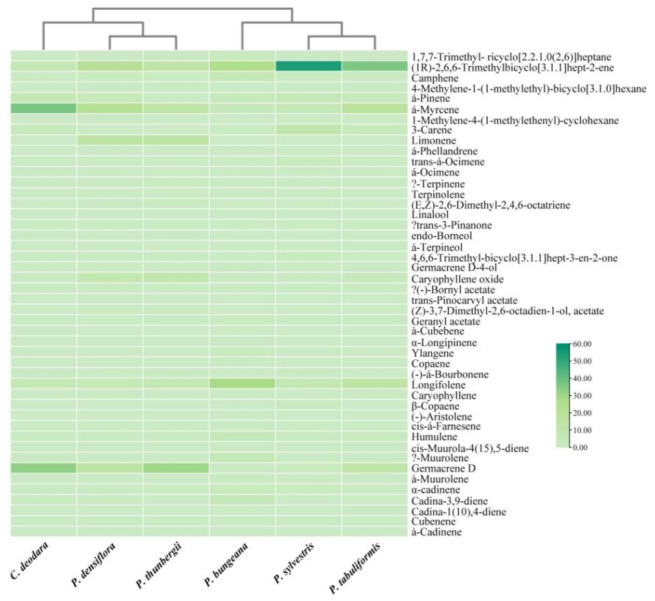
Cluster pedigree diagram of volatile terpenes and terpenoids in the pine leaf samples.

**Table 1 molecules-26-05244-t001:** Identification of volatile terpenes and terpenoids components and relative contents of compounds in pine leaves (Means ± SD).

NO.	Components	CAS No.	RI ^b^/RIL ^c^	Relative Content/%					
				*P. sylvestris*	*P. tabuliformis*	*P. bungeana*	*C. deodara*	*P. densiflora*	*P. thunbergii*
	Monoterpene								
1	1,7,7-Trimethyl- ricyclo [2.2.1.0(2,6)]heptane	508-32-7	924/927	0.25 ± 0.04	0.15 ± 0.04	2.25 ± 0.59	N.D. ^a^	0.27 ± 0.04	0.28 ± 0.05
2	(1R)-2,6,6-Trimethylbicyclo[3.1.1]hept-2-ene	7785-70-8	935/937	53.81 ± 7.01	36.20 ± 5.62	25.98 ± 1.23	6.53 ± 1.26	20.54 ± 2.21	11.72 ± 1.16
3	Camphene	79-92-5	950/950	1.33 ± 0.19	0.85 ± 0.17	5.39 ± 0.39	0.09 ± 0.03	1.07 ± 0.16	1.15 ± 0.20
4	4-Methylene-1-(1-methylethyl)-bicyclo[3.1.0]hexane	3387-41-5	974/974	N.D.	0.15 ± 0.01	N.D.	N.D.	0.43 ± 0.02	0.25 ± 0.01
5	á-Pinene	127-91-3	979/979	5.28 ± 0.49	3.23 ± 0.36	3.87 ± 0.29	7.25 ± 1.44	4.25 ± 0.49	N.D.
6	á-Myrcene	123-35-3	991/992	6.23 ± 0.15	17.54 ± 0.72	3.95 ± 0.17	36.79 ± 9.01	22.27 ± 1.90	13.02 ± 0.34
7	1-Methylene-4-(1-methylethenyl)-cyclohexane	499-97-8	1006/1005	N.D.	N.D.	N.D.	N.D.	0.28 ± 0.02	N.D.
8	3-Carene	13466-78-9	1011/1013	N.D.	0.06 ± 0.00	N.D.	N.D.	N.D.	N.D.
9	Limonene	138-86-3	1030/1031	10.89 ± 0.21	4.37 ± 0.18	1.12 ± 0.05	3.42 ± 0.90	N.D.	N.D.
10	á-Phellandrene	555-10-2	1032/1031	N.D.	N.D.	N.D.	N.D.	14.75 ± 1.32	14.25 ± 0.74
11	trans-á-Ocimene	3779-61-1	1038/1041	N.D.	N.D.	0.82 ± 0.02	N.D.	N.D.	N.D.
12	á-Ocimene	13877-91-3	1048/1053	2.46 ± 0.30	N.D.	N.D.	N.D.	N.D.	N.D.
13	ç-Terpinene	99-85-4	1061/1062	N.D.	0.05 ± 0.01	0.19 ± 0.01	N.D.	0.10 ± 0.01	0.10 ± 0.01
14	Terpinolene	586-62-9	1091/1089	0.32 ± 0.02	0.90 ± 0.04	0.14 ± 0.01	0.40 ± 0.10	0.56 ± 0.04	1.49 ± 0.07
15	(E,Z)-2,6-Dimethyl-2,4,6-octatriene	7216-56-0	1134/1131	N.D.	N.D.	0.13 ± 0.01	N.D.	N.D.	N.D.
	**Oxygenated Terpenes**								
16	Linalool	78-70-6	1103/1103	N.D.	N.D.	N.D.	N.D.	0.10 ± 0.00	1.62 ± 0.27
17	trans-3-Pinanone	547-60-4	1170/1170	0.07 ± 0.01	N.D.	0.10 ± 0.01	N.D.	N.D.	N.D.
18	endo-Borneol	507-70-0	1178/1172	N.D.	0.14 ± 0.04	N.D.	N.D.	0.28 ± 0.11	0.24 ± 0.11
19	à-Terpineol	98-55-5	1201/1198	N.D.	N.D.	N.D.	0.21 ± 0.13	N.D.	0.13 ± 0.04
20	4,6,6-Trimethyl-bicyclo[3.1.1]hept-3-en-2-one	80-57-9	1218/1217	N.D.	0.04 ± 0.01	N.D.	N.D.	N.D.	N.D.
21	Germacrene D-4-ol	198991-79-6	1595/1578	1.09 ± 0.79	N.D.	N.D.	0.42 ± 0.61	0.10 ± 0.05	0.13 ± 0.03
22	Caryophyllene oxide	1139-30-6	1605/1592	N.D.	0.07 ± 0.01	0.41 ± 0.08	N.D.	N.D.	N.D.
	**Terpene esters**								
23	(-)-Bornyl acetate	5655-61-8	1292/1289	0.91 ± 0.39	3.63 ± 1.42	N.D.	0.16 ± 0.02	8.69 ± 0.15	9.54 ± 0.63
24	trans-Pinocarvyl acetate	1686-15-3	1306/1298	N.D.	N.D.	N.D.	0.07 ± 0.01	N.D.	N.D.
25	(Z)-3,7-Dimethyl-2,6-octadien-1-ol, acetate	141-12-8	1364/1362	N.D.	N.D.	N.D.	N.D.	0.11 ± 0.03	N.D.
26	Geranyl acetate	105-87-3	1384/1379	N.D.	N.D.	N.D.	N.D.	1.07 ± 0.41	2.25 ± 0.57
	**Sesquiterpenes**								
27	à-Cubebene	17699-14-8	1356/1352	0.34 ± 0.15	0.07 ± 0.01	1.02 ± 0.12	0.10 ± 0.02	N.D.	0.16 ± 0.01
28	α-Longipinene	5989-08-2	1363/1358	N.D.	N.D.	N.D.	0.08 ± 0.01	N.D.	N.D.
29	Ylangene	14912-44-8	1381/1377	N.D.	N.D.	0.60 ± 0.08	N.D.	N.D.	N.D.
30	Copaene	3856-25-5	1388/1381	0.41 ± 0.23	N.D.	1.73 ± 0.18	0.28 ± 0.09	N.D.	N.D.
31	(-)-á-Bourbonene	5208-59-3	1398/1393	N.D.	N.D.	0.39 ± 0.03	N.D.	N.D.	N.D.
32	Longifolene	475-20-7	1422/1418	0.55 ± 0.24	N.D.	N.D.	0.13 ± 0.05	N.D.	0.83 ± 0.25
33	Caryophyllene	87-44-5	1434/1430	7.33 ± 2.01	13.98 ± 3.36	28.42 ± 0.75	6.20 ± 2.03	6.40 ± 1.81	8.56 ± 1.01
34	β-Copaene	18252-44-3	1435/1442	N.D.	0.32 ± 0.06	N.D.	0.00 ± 0.00	0.40 ± 0.12	N.D.
35	(-)-Aristolene	6831-16-9	1452/1455	2.19 ± 1.31	N.D.	N.D.	N.D.	N.D.	N.D.
36	cis-á-Farnesene	28973-97-9	1458/1458	N.D.	N.D.	0.19 ± 0.01	N.D.	N.D.	N.D.
37	Humulene	6753-98-6	1469/1468	1.11 ± 0.37	2.51 ± 0.52	5.82 ± 0.42	1.35 ± 0.49	1.24 ± 0.62	1.64 ± 0.15
38	cis-Muurola-4(15),5-diene	157477-72-0	1478/1469	0.11 ± 0.05	0.14 ± 0.02	0.13 ± 0.01	0.18 ± 0.05	0.16 ± 0.04	0.30 ± 0.03
39	ç-Muurolene	30021-74-0	1489/1480	0.52 ± 0.20	0.32 ± 0.04	6.49 ± 0.23	0.64 ± 0.21	0.36 ± 0.11	0.89 ± 0.02
40	Germacrene D	23986-74-5	1498/1489	2.30 ± 1.01	13.50 ± 2.25	0.23 ± 0.01	32.70 ± 9.78	14.86 ± 4.14	30.19 ± 1.16
41	à-Muurolene	10208-80-7	1513/1505	N.D.	0.32 ± 0.02	1.66 ± 0.09	0.39 ± 0.08	N.D.	N.D.
42	α-cadinene	39029-41-9	1528/1521	2.03 ± 0.92	0.48 ± 0.03	2.66 ± 0.15	0.83 ± 0.15	0.57 ± 0.11	0.99 ± 0.03
43	Cadina-3,9-diene	523-47-7	1534/1520	N.D.	N.D.	5.73 ± 0.40	1.43 ± 0.22	0.96 ± 0.12	N.D.
44	Cadina-1(10),4-diene	483-76-1	1536/1528	N.D.	0.81 ± 0.03	N.D.	N.D.	N.D.	N.D.
45	Cubenene	29837-12-5	1546/1535	0.12 ± 0.05	N.D.	0.27 ± 0.03	0.08 ± 0.02	N.D.	N.D.
46	à-Cadinene	24406-05-1	1551/1542	0.37 ± 0.28	0.16 ± 0.01	0.31 ± 0.03	0.28 ± 0.05	0.17 ± 0.03	0.25 ± 0.02

^a^ N.D. indicates not detected; ^b^ RI: Retention index determined experimentally on a DB-5MS capillary column (30 m × 0.25 mm; 0.25 μm film thickness); ^c^ RTL: Retention indices found in literature or database (NIST 14).

## Data Availability

Data in this article are available on request from the corresponding author.

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
