# Peer review of "Comparative Analysis of Volatile Terpenes and Terpenoids in the Leaves of *Pinus* Species—A Potentially Abundant Renewable Resource"

_molecules, 2021, doi:10.3390/molecules26175244_

Round 1
Reviewer 1 Report
Dear Editor and Authors,
Regarding the manuscript ID: molecules-1348624 entitled “Comparative Analysis of Volatile Terpenes and Terpenoids in the Leaves of Pinus Species, A Potentially Abundant Renewable Resource” my comments are the following.
The authors study the volatile terpenes and terpenoids in Pinus sp. leaves. For this purpose, they collected six (?) kinds of Pine sp. and directly determined the volatiles compounds in leaves without prior treatment.
The subject of the study is very interesting, the introduction section is very informative, giving a lot of information related to the topic and smoothly connects the issues under study. I believe that there is no need for extensive changes in the manuscript, but the discussion section needs to be compared or discussed with the existing literature. Finally, there is something that needs to be clarified: throughout the manuscript you are referring to 6 species of Pinus, we understand the same from the figures. However, in some parts you are talking about 10 species (see lines 21, 148, 174). Please explain and/or correct.
In addition, in some parts of the manuscript sometimes you refer to needles and sometimes to leaves. Please keep one of the two terms throughout the text.
Abstract line 15: I believe that the correct term is headspace solid-phase microextraction.
Based in the above I suggest minor revisions.
Reviewer 2 Report
The manuscript “Comparative Analysis of Volatile Terpenes and Terpenoids in the Leaves of Pinus Species, A Potentially Abundant Renewable Resource” deserves publication and the paper needs only minor changes.
Line 60 - “a large number of literature has 60 been reported on the volatile oil of pine leaves” – please provide sample literature references
Line 79 – IR – please define the abbreviation on first use
Lines 116-117 - the information analogous to lines 79-81 does not add anything new to the work.
Fig.1 and Fig.2 – TA and TN – please define the abbreviation
Fig.1 - no description in the legend what the blue dots mean
